# Quantitative Trait Locus Analysis of Hessian Fly Resistance in Soft Red Winter Wheat

**DOI:** 10.3390/genes14091812

**Published:** 2023-09-17

**Authors:** John W. Bagwell, Madhav Subedi, Suraj Sapkota, Benjamin Lopez, Bikash Ghimire, Zhenbang Chen, G. David Buntin, Bochra A. Bahri, Mohamed Mergoum

**Affiliations:** 1Institute of Plant Breeding, Genetics and Genomics, University of Georgia, Griffin Campus, Griffin, GA 30223, USA; jbagwell@uga.edu (J.W.B.); ms02824@uga.edu (M.S.); bikash.ghimire@uga.edu (B.G.); bbahri@uga.edu (B.A.B.); 2Small Grains and Potato Germplasm Research Unit, United States Department of Agriculture Agricultural Research Service, Aberdeen, ID 83210, USA; suraj.sapkota@usda.gov; 3Department of Crop and Soil Sciences, University of Georgia, Griffin Campus, Griffin, GA 30223, USA; benlopez@uga.edu (B.L.); zchen@uga.edu (Z.C.); 4Department of Plant Pathology, University of Georgia, Griffin Campus, Griffin, GA 30223, USA; 5Department of Entomology, University of Georgia, Griffin Campus, Griffin, GA 30223, USA; gbuntin@uga.edu

**Keywords:** wheat, hessian fly, quantitative trait locus, KASP, *H24*, *H32*

## Abstract

The Hessian fly (HF) is an invasive insect that has caused millions of dollars in yield losses to southeastern US wheat farms. Genetic resistance is the most sustainable solution to control HF. However, emerging biotypes are quickly overcoming resistance genes in the southeast; therefore, identifying novel sources of resistance is critical. The resistant line “UGA 111729” and susceptible variety “AGS 2038” were crossbred to generate a population of 225 recombinant inbred lines. This population was phenotyped in the growth chamber (GC) during 2019 and 2021 and in field (F) trials in Georgia during the 2021–2022 growing seasons. Visual scoring was utilized in GC studies. The percentage of infested tillers and number of pupae/larvae per tiller, and infested tiller per sample were measured in studies from 2021 to 2022. Averaging across all traits, a major QTL on chromosome 3D explained 42.27% (GC) and 10.43% (F) phenotypic variance within 9.86 centimorgans (cM). SNP marker *IWB65911* was associated with the quantitative trait locus (QTL) peak with logarithm of odds (LOD) values of 14.98 (F) and 62.22 (GC). *IWB65911* colocalized with resistance gene *H32*. KASP marker validation verified that UGA 111729 and KS89WGRC06 express *H32*. *IWB65911* may be used for marker-assisted selection.

## 1. Introduction

The Hessian fly (HF), or *Mayetiola destructor* Say [1], is one of the oldest recorded invasive species in North America. It can cause substantial economic damage wherever wheat is grown [2]. The US experienced major HF epidemics in the past, so the government started programs to control this catastrophic pest in large wheat-growing regions [1]. Sixteen million acres of HF-resistant wheat were planted nationwide in 1974 [1]. HF infestation usually lowers grain yield more than quality. If over five percent of tillers are infested during the early tillering stage, yield loss can be considered significant [3]. HF has caused millions of dollars in damage to US wheat. South Carolina lost approximately $4 million annually from 1984 to 1989, and Georgia lost roughly $20 million from 1988 to 1989 [4]. HF can cause annual GA field losses of up to 10% [5]. In Plains and Tifton, GA, single-stem samples of wheat averaged 1.97 larvae for each infested stem [5]. HF lowered the average grain weight of infested stems by 41.3% [5]. In Oklahoma, one immature HF per tiller can lead to approximately 31.27 kilograms (kg) per hectare (ha) in losses [4]. This pest can infest quack grass [6], barley, rye, and wheat [7]. 

HF adult females may lay 100–400 eggs on leaf adaxial surfaces for around three hours [8]. After three to four days, larvae emerge from the eggs at 20 °C [8] to crawl down to the closest node to feed for two to three weeks in the first and second instar stages [4]. Larvae are more likely to survive on younger leaves because there are more responsive cells that larvae can convert to galls for better nutrition [9]. Third-instar larvae pupate, and this stage lasts for 7–35 days [4]. However, pupae can remain dormant for three to four months in wheat stubble [10]. If temperatures stay at least 21 °C and humidity remains high for 10–14 days, adults usually emerge from pupae to mate and lay eggs. This temperature is ideal for HF to develop [4]. The larval stage is when damage is inflicted on wheat. As larvae feed, they turn the base of wheat plants into nutritive tissue, stunting their tiller growth [3]. They can also cause lodging, smaller wheat kernels and spikes, and fewer kernels per spike [1]. Other infestation symptoms include unusually short leaf blades, sheaths, and internodes, as well as darker green leaf color [3]. 

Resistant cultivars are the most cost-effective control option [3], especially in the Southeast (SE) of the US where fly-free dates are less effective [11]. The SE has a more optimal climate for HF to produce more generations than in other regions [11]. While 37 HF R genes were identified [12], only a few, such as *H13*, work well in the SE. Genes that used to confer higher resistance to HF in soft red winter wheat (SRWW) in the SE, including *H3*, *H5*, *H6*, *H7*, and *H8*, are no longer as effective [13]. *H9* is losing its efficacy in parts of the SE, and *H18* is temperature-sensitive [14]. Microsatellite data [15] and virulence assays [13] indicated that there is just one major population of HF in the SE US with population structure as well as microscale diversity [13,15]. HF populations from Holmes County, Mississippi, and Florence County, South Carolina, had a different population identity when compared to populations from other counties from SE states [16]. The *H12*, *H13*, *H18*, *H24*–*H26*, *H31*–*H33*, and *Hdic* genes are still effective in multiple SE counties, but *H24*–*H26* can be associated with undesirable agronomic traits [14]. Recently discovered quantitative trait locus (QTL), *QHft.nc-7D*, has been linked with partial field resistance in North Carolina (NC) [17]. More sources of resistant germplasm are needed to combat HF and avoid overcoming the few available effective R genes in the SE. 

Several diversity panels as well as biparental populations were developed to conduct GWAS and QTL analysis and identify genomic regions involved with HF resistance. A diversity panel of hard red spring (HRS), soft white spring (SWS), and soft white club spring (SWC) wheat evaluated for seedling HF resistance in Moscow, Idaho (ID), revealed *IWA6803*, a significant SNP closely linked to *H34* on chromosome 6B, and a novel QTL, *QHf.pnw.2B*, on chromosome 2B [18]. Winter wheat diversity panel AM203 in Manhattan, Kansas, was used to validate KASP-3B3797431 and KASP-3B4525164, which could be near diagnostic markers to detect *QHf.hwwg-3B*, a QTL on chromosome 3B mapped to 6.79 Mb, explaining up to 46.7% phenotypic variation (PV) for HF resistance [19]. Analysis from a diversity panel, biparental population, and elite ICARDA lines all of durum wheat revealed *QHara.icd-6B*, a locus explaining 83% PV with a 54.5 logarithm of odds (LOD) value that did not demonstrate yield drag when evaluated across locations [20]. 

Recombinant inbred line (RIL) population Seneca, developed from a cross between HF-resistant spring wheat variety Bobwhite and winter wheat variety Seneca (CI 12529), was used in conjunction with genotyping-by-sequencing (GBS) SNPs for mapping to reassign *H7*, a major gene explaining up to 78.3% PV, from chromosome 5D to 6A [21]. *H35* from chromosome 3BS and *H36* from chromosome 7AS were two HF resistance genes detected using a 154 RIL population generated from resistant HRWW line SD06165 and susceptible line OK05312. *H35* was a major QTL explaining up to 36% PV, and *H36* was a minor QTL explaining up to 13.1% PV [12]. The major QTL *QHf.wak-1A* was mapped in a registered spring wheat RIL population [22] produced from a cross between resistant line Louise and susceptible line Penawawa, which explained up to 90% PV for HF resistance [23]. 

Breeding efforts in the US have led to the release of several HF-resistant cultivars. In the SE region, resistance to HF is necessary, and breeding programs releasing cultivars for this region must incorporate genes for HF resistance in the newly developed cultivars. The UGA breeding program has released numerous cultivars adapted to the SE region with *H13* and *H9* genes that confer resistance to HF in GA and the SE. This includes recently released cultivars in 2020 (AGS 2021, PGX 20-15, and AP 1983) and in 2022 (AGS 3026, AGS 4023, and USG 3725) (Mergoum, Personal communication). *H24* is one of the R genes still highly effective against SE HF field collections [14]. HF resistance on wheat line KS89WGRC06 was deemed to be governed by *H24* on the long arm of chromosome 3D via monosomic analysis [24,25] and RFLP marker validation in the early 1990s. The *H24* linked RFLP markers were *XcnlBCD451*, *XcnlCDO482*, and *XksuG48* [26]. 

Chromosome 3D also has R genes *H26* and *H32*, derived from wheat lines KSWRCG26 and W-7984, respectively [14]. Seedlings with single R genes *H24*, *H26*, or *H32* exposed to HF populations from AL, GA, and NC demonstrated 75–100%, 87.8–100%, and 83.2–99.5% resistance, respectively [14]. *H32* has been mapped in between flanking simple-sequence repeat (Xgwm3 and Xcfd223) [27], sequence-tagged site (*Xrwgs10* and *Xrwgs12*) [28], and SNP markers (*IWB65911* and *IWB37580*) [29]. *Xrwgs10* and *Xrwgs12* are also tightly linked to *H26* [30], so *H26* and *H32* can be easily introgressed together [10]. Since *H24* and *H26* can be associated with unideal agronomic traits due to linkage drag, *H32* may be an alternative that does not lower yield as much [14]. 

Kompetitive Allele-Specific PCR (KASP) markers were created for some R genes to quickly, cheaply, and accurately screen cultivars and accelerate marker-assisted selection (MAS) for plant breeding. KASP primer sets were developed for SNP *IWB65911* that cosegregates with *H32*, and it has differentiated HF-susceptible cultivars from resistant ones with high sensitivity as well as specificity [29]. R genes *h4*, *H7*, *H35*, and *H36* also have KASP markers, but validation is needed before using these markers for MAS [10]. KASP-6B7901215 and KASP-6B112698 were validated to deploy *QHf.hwwg-6BS*, a major QTL explaining up to 84% PV that was derived from the cultivar Chokwang [31]. Using KASP markers and other techniques, such as crossing durum wheat to bread wheat and doubling F1 chromosomes via colchicine, can expedite the introgression of new HF R genes [10,32]. 

UGA 111729 and AGS 2038 are elite SRWW breeding lines developed by the University of Georgia (UGA) Small Grains Breeding Program (UGA-SGBP). AGS 2038 was developed from a cross between Pioneer 26R61 and GA 961581, and it was released to AGSouth Genetics in 2011 [33]. Pioneer 26R61 has HF resistance QTL *QHf.uga-2AS*, *QHf.uga-3DL*, and *QHf.uga-6AL* on chromosomes 2A, 3DL, and 6A, respectively. *QHf.uga-6AL* is a major QTL flanked by SSR marker *Xgwm427* and DArT marker wPt-731936 that explained up to 63% average PV. *QHf.uga-2AS* is a minor QTL flanked by SSR markers *Xgwm359* and *Xbarc124*. *QHf.uga-3DL* was considered different from *H24*, *H26*, and *H32* and significant during the late seedling growth stage since PV and LOD values increased as seedlings aged [34]. UGA 111729 was developed by backcrossing KS89WGRC06 to AGS 2038. Thus, it is presumed to carry *H24*. These two cultivars were crossed to develop a biparental RIL population that was first studied for leaf rust resistance [33].

Therefore, in this study, the objectives were to identify genomic regions involved with HF resistance using the SRWW UGA 111729 × AGS 2038 biparental RIL population for QTL analysis and to determine the most significant marker intervals influencing HF resistance that could be used for MAS.

## 2. Materials and Methods

### 2.1. Plant Materials

A biparental RIL F_6_ population of 225 lines derived from a cross between HF-susceptible parent AGS 2038 and HF-resistant parent UGA 111729 was used in this study along with cultivars AGS 3030 as a resistant check [35] and USG 3555 as a susceptible check [36]. UGA 111729 is thought to have HF R gene *H24* because its progenitor, KS89WGRC06, carries it [25,26]. The UGA-SGBP developed all these genotypes except USG 3555. While UGA 111729 is an elite line, AGS 3030 (GAJT141-14E45) and AGS 2038 (GA001138-8E36) were released by UGA in 2017 [35] and 2011 [33], respectively, and were both licensed to AgSouth Genetics [33,35]. USG 3555 (VA02W-555) (PI 654454) is an SRWW line developed by the Virginia Agricultural Experiment Station and released in 2007 [36]. This population was genotyped, and its linkage map was constructed, as described by Sapkota et al. [33], with a 90K SNP array and 8,800 selected polymorphic markers.

### 2.2. Field Experimental Design

Field experiments were conducted at the University of Georgia (UGA) Southwestern Research and Education Center (SWREC) in Plains, GA (32.04723600057329° N, −84.36617512994249° W), and Bledsoe Research Farm in Williamson, GA (33.173149812149354° N, −84.40675154213565° W), during the 2021 growing season with two replicates. For the 2022 growing season, one replicate was planted in Williamson for QTL validation. Two checks were added to the study, and a check was replicated and distributed for every 20 rows in the study. Including RILs and parents, each block had 237 lines. A randomized complete block design (RCBD) was implemented for each field with three filler rows of variety USG 3555 at the end of each block (Appendix A). The UGA SWREC field plot has Greenville sandy clay loam soil [37], and the Bledsoe field plot has Cecil sandy loam soil [38]. In 2020, seed was planted in Williamson on November 6 and in Plains on November 20. In 2021, seed was planted in Williamson on October 27. The fields were irrigated to ensure adequate and consistent germination across the field. Each field alleyway had a 1.5 m spacing. One row was 1 m long. Any four rows planted together had 30 cm of space between each other. Natural HF damage was relied upon for the plots. Susceptible wheat variety USG 3555 used as HF trap crops were planted around the experimental fields at both locations in late August of each year (Mergoum Lab, Personal communication).

A pre-plant fertilizer, including Nitrogen (N) at 22.97 kg of urea N per ha, 20.17 kg of phosphorus (P), and 57.15 kg of potassium (K), was applied at 459.46 kg/ha in October–November each year. Zidua, a granular pre-emergence herbicide, and ProwlH_2_O were applied at 46.23 g/ha and 2.24 kg/ha after planting in early November, respectively. In early February, usually the wet time of the year, both plots were top-dressed with 36.28 kg of urea ammonium nitrate (UAN) in liquid form at 211.37 L/ha. Harmony Extra, a broad-leaf herbicide, was applied around the same time at 146.15 mL/ha.

### 2.3. Growth Chamber Experimental Design

Growth chamber experiments were conducted at UGA, Griffin Campus (33.26445975342215° N, −84.28409533175976° W). The experiments included the same lines studied in the field, arranged in an RCBD, in two and three replicates in March 2019 and October 2021, respectively [39]. Seeds were planted in cones in cone-tainer trays (Stuewe and Sons, Inc., Tangent, OR). Each cone-tainer tray was 30.48 cm × 60.96 cm × 17.15 cm and held 98 cones, with each cone being 3.81 cm in diameter at the largest point and approximately 10.16 cm tall. The cones were filled with Pro-Mix growing mix (Pro-Mix Gardening, Quakertown, PA) and subjected to 14 h days, 10 h nights, and 18 ± 3 °C in the greenhouse [12]. Three seeds per line were planted in each cone. Once the first true leaf expanded and the second leaf emerged from each seedling (Feekes stage 1) [40], the cone-tainers were moved to a PGR15 growth chamber (Conviron, Pembina, ND), and third-instar HF larvae of biotype L obtained from Purdue University (40.4235665310146° N, −86.92151646777332° W) were put in metal pans beside the cone-tainers. The larvae pupated, and adult HF emerged approximately a couple of days after they arrived at their destination. Stored adults were not sent because they had lower fecundity than third-instar larvae (Cambron, Personal communication). An amount of 1050–1500 pupae was used to induce high infestation pressure for each growth chamber experiment. Growth chamber conditions included a 14 h photoperiod with 20 °C, 875 μMol, and level 4 incandescent light intensity during the day and no light and 15 °C at night. These conditions were used to simulate field conditions for the seedlings as closely as possible (Mergoum Lab, Personal Communication).

### 2.4. Data Collection 

Field data collection was conducted at Feekes stage 10.5 [40]. We sampled 20 plants from each entry and block. In 2021 in Plains, the two replicates were collected in April. All entries from Williamson were sampled in June 2021 and May 2022. The adult samples were then scored for HF larvae and pupae infestation by averaging the HF count of the 20 plants from each sample, also known as the number of larvae/pupae (NOP) [17]. Percent infested tillers per sample (PIT), a trait describing pest instance in a sample, indicates how many tillers were infested by at least one HF larva. Based on PIT per sample, if the susceptible check and susceptible parent have less than 40–50% PIT, this threshold scale is used as follows: resistant = 0–10%, intermediate = 11–20%, and susceptible ≥20%. If susceptible checks and parents have at least 40–50% PIT, this scale is used as follows: resistant = 0–10%, moderately resistant = 11–20%, moderately susceptible = 21–30%, and susceptible ≥30% (Buntin, Personal communication). The number of larvae or pupae per tiller is NOP divided by the total number of tillers in a sample (NOPPT). Number of larvae or pupae per infested tiller is the total number of pupae or larvae divided by the number of infested tillers per line (NOPIT). NOPPT and NOPIT are traits that reveal pest severity in a line [17]. For NOPPT and NOPIT, RILs were assigned to class resistant, intermediate, or susceptible by comparing their values to their parents. The NOPPT scale for Plains and Williamson was resistant = 0–<0.5625, intermediate = 0.5625–<1.125, and susceptible ≥ 1.125; the NOPIT scale was resistant = 0–<1.807, intermediate = 1.807–<3.614, and susceptible ≥ 3.614 (Bahri, Personal communication). 

Growth chamber data were collected three weeks after infestation from HF. Data were collected by scoring plant morphology in both the 2019 and 2021 experiments. Stunted, dark green seedlings were counted as susceptible, while non-stunted, light green plants were counted as resistant [12]. The percentage of resistant plants (Res) for each line was recorded for QTL analysis [12]. Res was the only trait documented in the 2019 experiment. In the 2021 experiment, PIT, NOPPT, NOPIT, and Res were evaluated for each line. RILs were assigned to class resistant, intermediate, or susceptible for PIT, NOPPT, and NOPIT by comparing their values to their parents and the checks. The PIT scale was resistant = 0–<48.15%, intermediate = 48.15–<70.37%, and susceptible = 70.37–100%; the NOPPT scale was resistant = 0–<2.96, intermediate = 2.96–<5.07, and susceptible ≥ 5.07; and the NOPIT scale was resistant = 0–<1.56, intermediate = 1.56–<2.01, and susceptible ≥ 2.01 (Bahri, Personal communication).

### 2.5. Phenotypic Data Analysis 

Phenotypic analyses were conducted in R version 4.2.2 (Posit Software, Boston, MA) and Microsoft Excel. Type I analysis of variance (ANOVA), Chi-square (X^2^), and Pearson correlation analyses were conducted in base R. The significance level for each individual Pearson correlation was computed using R package Hmisc [41], and Type II and III ANOVA were conducted using R package car [42]. Correlations and their statistical significance were visualized together using Microsoft Excel. ANOVA was conducted to check for statistical differences between means of RILs. Type I ANOVA was used for datasets with no missing values and when no interactions between variables were statistically significant; Type II ANOVA was used for datasets with missing values and no statistically significant interactions between variables; and Type III ANOVA was used for datasets with missing values and statistically significant interactions. X^2^ analysis was conducted to compare differences between observed and expected results for all traits and if the differences could be due to single-gene or multiple-gene influence. Correlation analysis was conducted to detect linear relationships among all variables. Frequency distributions were generated in Microsoft Excel to visualize the segregation of HF resistance or susceptibility in the RIL population. The Shapiro–Wilk test was used to check the data for normality. Levene’s test helped test data for homogeneity [43]. 

Broad-sense heritability (*H^2^*) was calculated using the R package lme4 [44]. If the field data were homogeneous, *H^2^* for field plots was calculated with the following equation [33]
(1)$$H^{2}=\frac{\sigma_{G}^{2}}{\sigma_{G}^{2}+\frac{\sigma_{GE}^{2}}{r}+\frac{\sigma_{GY}^{2}}{r}+\frac{\sigma_{GEY}^{2}}{r}+\frac{\sigma_{e}^{2}}{eyr}\;}$$
where *σ^2^_G_* = genotypic variance, *σ^2^_GE_
*= genotype by environment (G × E) interaction variance, *σ^2^_GY_* is genotype by year variance, *σ^2^_GEY_* = G × E by year interaction variance, *σ^2^_e_* = error variance, *G* = genotype, *e* = environment, *y* = year, *r* = replicate, and *e* = error. If the data were not homogeneous, this *H^2^* formula was used instead [33]
(2)$$H^{2}=\frac{\sigma_{G}^{2}}{\sigma_{G}^{2}+\frac{\sigma_{e}^{2}}{r}\;}$$

If greenhouse data were homogeneous across years, *H^2^* for greenhouse studies was calculated with the following equation [33]
(3)$$H^{2}=\frac{\sigma_{G}^{2}}{\sigma_{G}^{2}+\frac{\sigma_{GY}^{2}}{r}+\frac{\sigma_{e}^{2}}{yr}\;}$$
where *σ^2^_G_* = genotypic variance, *σ^2^_GY_* is genotype by year interaction variance, *σ^2^_e_* = error variance, *G* = genotype, *y* = year, *r* = replicate, and *e* = error. If greenhouse data were heterogeneous, Equation (2) was used. There is no G × E interaction component here because important environmental factors can be controlled in a greenhouse environment. Narrow sense heritability (*h^2^*) was calculated using the R package rrBLUP [45]. This formula for *h^2^* was used if field data were homogeneous [33] as follows: (4)$$h^{2}=\frac{\sigma_{A}^{2}}{\sigma_{G}^{2}+\frac{\sigma_{GE}^{2}}{r}+\frac{\sigma_{GY}^{2}}{r}+\frac{\sigma_{GEY}^{2}}{r}+\frac{\sigma_{e}^{2}}{eyr}\;}$$

For *h^2^*, additive genetic variance (*σ^2^_A_*) replaced genetic variance *σ^2^_G_* in the numerator for Equations (4)–(6). If the data were not homogeneous, this *h^2^* formula was used instead [33] as follows:(5)$$h^{2}=\frac{\sigma_{A}^{2}}{\sigma_{G}^{2}+\frac{\sigma_{e}^{2}}{r}\;}$$

If greenhouse data were homogeneous across years, *h^2^* was calculated with this equation [33] as follows:(6)$$h^{2}=\frac{\sigma_{A}^{2}}{\sigma_{G}^{2}+\frac{\sigma_{GY}^{2}}{r}+\frac{\sigma_{e}^{2}}{yr}\;}$$

If greenhouse data were heterogeneous, Equation (5) was used. 

### 2.6. QTL Analysis, Candidate Gene Identification, and Linkage Disequilibrium Decay Analysis

QTL analysis for traits PIT, NOPPT, NOPIT, and Res for each block, the average between blocks, location, and year was conducted using the QTL IciMapping BIP function for bi-parental populations [46]. QTL was detected with 1 cM walk speed, 0.001 stepwise regression probability, 1,000 permutations to make LOD thresholds, type 1 error of 0.05, and the inclusive composite interval mapping of additive QTL method [33]. R package LinkageMapView was used to visualize linkage maps [47]. 

QTL was significant if SNP peaks surpassed the LOD threshold calculated by the permutation analysis, pairwise SNP estimates surpassed the half LD decay critical value (*r^2^* > 0.24), and these pairwise estimates were within the map distance for half LD decay. After detecting SNP markers flanking novel QTL, GrainGenes (https://wheat.pw.usda.gov/GG3/ (accessed on 10 July 2023)) and the literature were searched using the names of the flanking SNP markers. Linkage disequilibrium (LD) for linkage groups with significant QTL, 3A1 and 3D, were analyzed using TASSEL [48] and visualized using base R. LD parameters in TASSEL were set to a sliding window size of 50 [49]. LD decay was plotted over cM distance according to Hill and Weir [50]. Half the maximum LD decay, where *r^2^* and the locally weighted polynomial regression (LOESS) curve intersect, was considered the *r^2^* critical value [51].

### 2.7. KASP Validation of H32

KASP markers already developed for *H32* were used for validation in Spring 2023. Marker information was obtained from Tan et al. [29]. Primers were ordered from Eurofins Genomics LLC. Genomic DNA from parent lines, KS89WGRC06, a synthetic *H32* line, and 29 RILs that were either consistently resistant or susceptible were extracted using a modified cetyltrimethylammonium bromide (CTAB) protocol [52] and then diluted to 50 ng/μL [33]. Chosen RILs were either consistently resistant (RILs 5, 63, 79, 122, 146, 148, 202, and 218) or susceptible (RILs 7, 23, 26, 29, 31, 96, 101, 108, 113, 114, 174, 183, 184, 185, 205, 206, 209, 211, 213, 223, and 225) across years for greenhouse experiments for Res. KS89WGRC06 and the synthetic *H32* line were included in the KASP study as resistant checks. The following PCR program was used for KASP marker validation: 30 °C for 1 min during the pre-read stage; 95 °C for 10 min during the preheating stage, followed by a touchdown program of 10 denaturation cycles at 95 °C for 20 s; annealing/extension at 61 °C for 1 min with a 0.6-degree reduction every cycle; plus 30 cycles of 95 °C for 29 s and 56 °C for 1 min. Temperature was reduced to 30 °C for 1 min and 30 s during the post-read stage. HEX dye was used for the resistant allele, and FAM dye was used for the susceptible allele. Fluorescent signals were collected at the pre-read and post-read stages at 30 °C (Chen, Personal communication). Results were visualized by plotting levels of expression of the resistant allele against levels of expression of the susceptible allele. Phenotypic validation in a growth chamber was conducted simultaneously. AGS 3030 was included as a resistant check, and USG 3555 was included as a susceptible check. All lines included in the KASP marker study except KS89WGRC06 and the synthetic *H32* line were phenotypically validated.

## 3. Results

### 3.1. Phenotype Results and Frequency Distributions

Frequency distributions were drawn for all traits at each location and year. Overall, on average, HF-resistant parent UGA 111729 was 97.42% more resistant to HF than HF-susceptible parent AGS 2038 in Plains, while both parents demonstrated a similar response to HF in Williamson. In Plains, UGA 111729 had an average of 0.01 NOPPT (Figure 1a), 0.83 PIT (Figure 1b), and 0.17 NOPIT (Figure 1c), while AGS 2038 had an average of 1.13 NOPPT (Figure 1a), 35 PIT (Figure 1b), and 3.61 NOPIT (Figure 1c). Averaging across both years in Williamson, UGA 111729 had an average of 0 for NOPPT (Figure 2a,b), PIT (Figure 2c,d), and NOPIT (Figure 2e,f, Table 1), while AGS 2038 had an average of 0.13 NOPPT (Figure 2a,b), 4.17 PIT (Figure 2c,d), and 1.5 NOPIT (Figure 2e,f, Table 1). The parents likely had minor differences between the measured traits in Williamson compared to Plains due to lower insect pressure. On average, for growth chamber data, UGA 111729 was 46.03% more resistant to HF than AGS 2038. For 2019, UGA 111729 had an average of 100% Res, and AGS 2038 had an average of 0% Res (Figure 3a). For 2021, UGA 111729 had an average of 62.96% Res (Figure 3b), 2.96 NOPPT (Figure 3c), 48.15 PIT (Figure 3d), and 1.56 NOPIT (Figure 3e), and AGS 2038 had an average of 7.41% Res (Figure 3b), 5.07 NOPPT (Figure 3c), 70.37 PIT (Figure 3d), and 2.01 NOPIT (Table 1; Figure 3e).

Given that resistant parents and checks were mainly below 40% PIT in the field, the following susceptibility thresholds were given to RILs: resistant with PIT at most 10%; intermediate with PIT >10–20%; and susceptible with PIT >20% (Buntin, Personal communication). Susceptibility thresholds were confirmed arbitrarily for Res from the growth chamber studies by comparing RIL data to the parents and checks. Susceptible was 0% Res; moderately susceptible was >0–50% Res; moderately resistant was >50–80% Res; and resistant was >80% Res [21,31]. 

In Plains, 85.8%, 5.8%, and 8.4% of RILs were considered resistant, intermediate in resistance, and susceptible for PIT, respectively (Figure 1b); 94.7%, 4.4%, and 0.9% were resistant, intermediate in resistance, and susceptible for NOPPT, respectively (Figure 1a); and 88%, 12%, and 0% were resistant, intermediate in resistance, and susceptible for NOPIT, respectively (Figure 1c). In Williamson, averaging across years, 94.4%, 4.7%, and 0.9% were considered resistant, intermediate in resistance, and susceptible for PIT, respectively (Figure 2c,d); 99.3%, 0.2%, and 0.4% were considered resistant, intermediate in resistance, and susceptible for NOPPT, respectively (Figure 2a,b); and 98.4%, 1.3%, and 0.2% were considered resistant, intermediate in resistance, and susceptible for NOPIT, respectively (Figure 2e,f). In March 2019, for growth chamber data, 24.9% and 75.1% were considered resistant and susceptible for Res, respectively (Figure 3a). In October 2021, for growth chamber data, 40% and 60% were considered resistant and susceptible for Res, respectively (Figure 3b); 58.2%, 21.3%, and 20.4% were considered resistant, intermediate in resistance, and susceptible for PIT, respectively (Figure 3d); 63.1%, 22.2%, and 14.7% were considered resistant, intermediate in resistance, and susceptible for NOPPT, respectively (Figure 3c); and 65.8%, 12.4%, and 21.8% were considered resistant, intermediate in resistance, and susceptible for NOPIT, respectively (Figure 3e). 

### 3.2. X^2^ Tests, Normality Tests, and Heritability

For Res growth chamber data, to test for single gene segregation ratio, susceptible and moderately susceptible and resistant and moderately resistant were combined into susceptible and resistant, respectively. X^2^ results were calculated for all traits for all datasets. For all replicates and averages, while X^2^ values for both segregation ratio options (1:1 and 1:1:1:1) succeeded their respective critical values, single gene segregation had smaller values than multiple gene segregation (X^2^_(0.01,1)_ = 0.05 < 6.63, *p* > 0.01; X^2^_(0.01,2)_ = 0.05 < 9.210, *p* > 0.01; X^2^_(0.01,3)_ = 0.05 < 11.345, *p* > 0.01) (Table 2). For PIT, NOPPT, and NOPIT field and growth chamber data, all X^2^ values were larger than their corresponding critical values. Growth chamber X^2^ values were consistently the smallest, and Williamson X^2^ values were the largest (Table 3). 

All data were tested for normality utilizing the Shapiro–Wilk test and homogeneity with Levene’s test. All the data deviated significantly from a normal distribution and were heterogeneous. Thus, *H^2^* and *h^2^* were calculated separately for each dataset, not across environments or years. Equations (2) and (5) were used for *H^2^* and *h^2^*, respectively. For Plains, NOPPT and PIT had higher *H^2^* than NOPIT (0.60 vs. 0.64 vs. 0.29) as well as *h^2^* (0.47 vs. 0.52 vs. 0.18). Traits from Plains had higher *H^2^* (NOPPT: 0.60 vs. 0.12, PIT: 0.64 vs. 0.16, NOPIT: 0.29 vs. 0.19) and *h^2^* (NOPPT: 0.47 vs. 0.03, PIT: 0.52 vs. 0.03, NOPIT: 0.18 vs. 0.06) than Williamson. For growth chamber data, Res and PIT *H^2^* were higher than NOPPT and NOPIT (0.88 vs. 0.70 vs. 0.47 vs. 0.39) (Table 4).

### 3.3. Correlation between Phenotypic Traits and ANOVA Results

Overall, for trait averages, 77 statistically significant (*p* < 0.05) correlations were recorded across traits, environments, and years. The Pearson correlation coefficient ranged from −0.84 to 0.95. The highest correlation, 0.95, was observed for PIT vs. NOPPT in Plains 2021, while the lowest correlation, −0.84, was observed for PIT vs. Res for October 2021 growth chamber results (Table 5). Res was negatively correlated with PIT, NOPPT, and NOPIT (Appendix A). 

For ANOVA across both fields for 2021, line and location were significant factors for all traits. However, the line and location interaction were only significant for PIT and NOPPT. The line was significant for all measured traits for Plains, and nothing was significant for Williamson in either year. For growth chamber data, the line was significant for all traits for all datasets; replicate was significant for NOPPT and NOPIT in 2021; and Res had replicate as significant in 2019. Traits measured from the 2021 growth chamber trials were the only ones analyzed with Type I ANOVA because those were the only traits with no missing data (Appendix A). 

### 3.4. Quantitative Trait Locus and Linkage Disequilibrium Decay Analysis and Candidate Gene Identification

PIT, NOPPT, and NOPIT measured in the growth chamber had higher LOD and percent PV values than in the field. Res had the highest PV and LOD scores overall (LOD = 37.65, PV = 67.48%). NOPIT had the lowest PV and LOD scores for growth chamber results, and NOPPT had the lowest PV and LOD scores for field notes. Averaging across every trait, a major QTL within a 0.633 cM region on the long arm of chromosome 3D explained 10.43% and 37.92% PV in Plains and the growth chamber, respectively (Table 6 and Appendix A). No significant QTL was detected for Williamson during either year, but QTL peaks for all traits measured were detected in the same genetic location in Williamson in 2022 on the long arm of chromosome 3D. These peaks were just under the calculated LOD thresholds (Appendix A).

SNP markers *IWB65911* and *IWB37580* were associated with the QTL peak with LOD values of 14.98 and 37.54 for field and growth chamber data, respectively (Figure 4). One QTL detected on chromosome 6B flanked by SNP markers *IWB62788* and *IWB59262* within a region of 5.085 cM from 2019 Res data had a LOD value of 36.40 and explained 26.16% PV. Two minor QTLs were detected on chromosome 3A. The most distal one, flanked by *IWA6387* and *IWB62332* within a region of 4.992 cM, was from NOPPT from Plains 2021 data with a LOD value of 2.91 and 2.7% PV. The least distal one, flanked by *IWB72257* and *IWB14875* within a region of 5.2 cM, was from NOPIT from October 2021 growth chamber data with a LOD value of 4.12 and 5.9% PV (Table 6 and Appendix A).

Across the entire genome for the RIL progenies, LD decay dropped halfway at 30 cM. Thus, the population has large amounts of recombination, and most markers in the genome are unlinked (Appendix A). For individual genomes, half LD decay distance was the longest for the B genome (81 cM), while the D genome declined the fastest (14 cM). For individual linkage groups, the slowest LD decline was in linkage group 2D2 (160 cM), while the fastest LD decline was in linkage group 3B (1 cM) (Appendix A). Given the *r^2^* critical value of 0.24, or half the maximum LD decay, all statistically significant loci from linkage group 3A1 were linked (Figure 5a), and all statistically significant loci from chromosome 3D were linked (Figure 5b). The significant loci from linkage group 3A1 were named under QTL *QHf.ga.srww.3A*; the significant loci from chromosome 3D were named under QTL *QHf.ga.srww.3DL*; and the significant loci from linkage group 6B2 were named under QTL *QHf.ga.srww.6B*. 

### 3.5. KASP Marker and Phenotypic Validation

*IWB65911* was chosen for KASP marker validation because it was related to our significant QTL peak on the long arm of chromosome 3D. Also, it was already tested in a previous study (Appendix A) [29]. AGS 3030 and USG 3555 were not validated with the KASP marker. Res was used to compare lines for this validation study. The synthetic *H32* line, KS89WGRC06, and UGA 111729 all expressed the homozygous R allele, meaning that the allele should be associated with resistance to HF. AGS 2038 expressed the homozygous S allele, meaning that the allele should be associated with susceptibility to HF (Figure 6). A percentage of 96.3% of homozygous lines tested with *IWB65911* either had a consistent resistant phenotype and genotype or a consistent susceptible phenotype and genotype (Appendix A).

## 4. Discussion

HF is a highly damaging insect species to wheat in the US SE. Only six R genes are highly effective for that region, and three of them may lower agronomic traits [14]. HF can easily overcome these genes under high host selection pressure for HF virulence [32]. HF biotypes are constantly evolving and overcoming introduced R genes. Biotype L HF is currently the dominant biotype in the US SE [17]. While HF R gene *H13* is still effective against it [53], biotype *vH13* is overcoming this R gene, necessitating a search for novel resistance [17]. HF-resistant cultivars were demonstrated during an infestation to save $100/ha–$240/ha in damages compared to HF-susceptible cultivars [54], so finding novel HF resistance benefits farmers.

In this study, an SRWW biparental population was used to identify genomic regions involved with HF resistance. Yearly Res X^2^ values ratio were lower for a 1:1 segregation than a 1:1:1:1 ratio, meaning observed Res values were more likely to fit 1:1 ratio X^2^. This result means that one major QTL expressing HF resistance in UGA 111729 is more likely than multiple QTL. This is similar to X^2^ results from Zhang et al. [31], which found a major QTL that explained resistance to HF in the cultivar Chokwang. 

Our QTL results and KASP and phenotypic validations revealed one major gene for HF resistance in 3DL associated with the *IWB65911* marker. The HF resistance in UGA 111729 was detected both in the growth chamber and field trials, suggesting it is expressed from the seedling to the adult stage. *IWB65911* was used for previous KASP marker validation for HF studies, and they co-segregated with *H32* [29]. This finding is interesting, considering that UGA 111729 is supposed to carry *H24*, inherited from its progenitor KS89WGRC06. *H24* and *H32* are at least 20 cM away from each other [27]. Since Tan et al. [29] reported that *IWB65911* demonstrated a specificity of 1 and sensitivity of 0.93–0.94, our findings validate the efficacy of *IWB65911* due to our 96.3% result. Since *H24* does not currently have a publicly available SNP marker, RFLP marker validation with flanking markers, *Xcdo428* and *Xbcd451*, would need to be conducted to confirm that UGA 111729 has *H24* [26]. Since *Xrwgs10* and *Xrwgs12* are linked to *H26* as well as *H32*, STS marker validation can be used to determine if UGA 111729 also has *H26* [28]. 

As for our candidate genes from significant QTL, *IWA6387*, associated with *QHf.ga.srww.3A*, was a flanking SNP with SSR marker *Xbarc12* as part of an additive QTL, *QShi.hwwgr-3AS*, that explained up to 5.6% PV for wheat grain quality in a F_10–12_ RIL winter wheat population [55]. For *QHF.ga.srww.6B*, no candidate genes were found for SNPs *IWB62788* and *IWB59262*. *QHf.ga.srww.6B* was a major growth chamber QTL in this study; however, it was only detected in one replicate for 2019 results and not for 2021. *QHf.ga.srww.6B* should be further investigated.

Data from Plains had higher heritability than Williamson, indicating Williamson had higher environmental variance. Res and PIT *H^2^* were higher than all other traits for growth chamber data, meaning these two traits are more replicable than the other traits. In Plains, genetic causes from multiple genes can explain NOPPT and PIT better than NOPIT since NOPPT and PIT had higher *H^2^* and *h^2^*. Winn et al. [17] had a similar observation and suggested that PIT and NOPPT continue to be used to assess pest instance and pest severity, respectively. This study is the first to assess the correlations between PIT, NOPPT, NOPIT, and Res for HF resistance. All Res replicates and averages across years were negatively correlated with all other trait replicates and averages across years. This is expected since higher values for Res mean higher resistance vs. lower resistance for those higher values for PIT, NOPPT, and NOPIT. Res was the most strongly correlated with PIT across the years. When looking at correlations for averages within each individual year, PIT vs. NOPPT consistently had stronger correlations than PIT vs. NOPIT. PIT is likely to help determine how high or low NOPPT values will be. PIT should be the priority trait when phenotyping because it is easier to assess than NOPPT and NOPIT and it is highly correlated with NOPPT and Res.

The parents had smaller differences between the measured traits in Williamson vs. Plains due to lower insect pressure. There was evident G × E interaction between Plains and Williamson field results. One explanation for this G × E interaction and differences in insect pressure could be the number of acres planted in Pike County and Sumter County, where Williamson and Plains are located, respectively. The most recent publicly available data on acres harvested for individual Georgia counties dates to 2017. For 2012, in which there is data for both counties, Sumter County harvested 11,133 acres of wheat, and Pike County harvested 492 acres of wheat. In 2017, Sumter County harvested 2,523 acres, while there is no available information for Pike County (https://www.nass.usda.gov (accessed on 9 August 2023)). HF are more likely to reproduce in areas with more wheat planted and warmer climates (Mergoum Lab, Personal communication) [3,4]. There could have also been a difference in biotype composition per county, considering that lines with *H32* were shown to be more effective against Sumter County biotypes than Tift County biotypes [13]. However, Cambron et al. [13] did not have any results for the effect of Pike County biotypes on lines with *H32*.

There is not much literature to directly compare our LD decay results among RIL populations for HF response QTL. Bassi et al. [20] studied durum wheat, and Ando et al. [18] and Joukhadar et al. [56] used bread wheat diversity panels, but they did not compare subgenome or individual linkage group LD decay. Also, these studies did not assess PIT, NOPPT, or NOPIT. Pariyar et al. [57] used LD decay analysis with GWAS, considered 0.1 as their *r^2^* critical value, and had LD decay values of 2 cM and 6 cM for chromosomes 3A and 3D, respectively. Although their 3A LD decay value was smaller than ours at 23 cM, our 3D LD decay value was equal to theirs [57]. 

*H32* is still effective against HF biotypes in the US SE [14], which was also confirmed in our study. Despite KS89WGRC06 (known to carry *H24*) being a progenitor of UGA 111729, this paper validated the presence of *H32* in UGA 111729. This novel finding is valuable, considering that a KASP marker was developed for *H32* detection. Since the US SE is losing effective HF R genes, *IWB65911* should be used for MAS to introgress *H32* into new varieties for HF resistance, and the effect of *H32* on yield should be evaluated. *H32* should also be pyramided with other HF R genes for better resistance management against quickly evolving biotypes. This study demonstrates the efficacy of *QHf.ga.srww.3DL* and that breeders can use *IWB65911* for MAS. 

## Figures and Tables

**Figure 1 genes-14-01812-f001:**
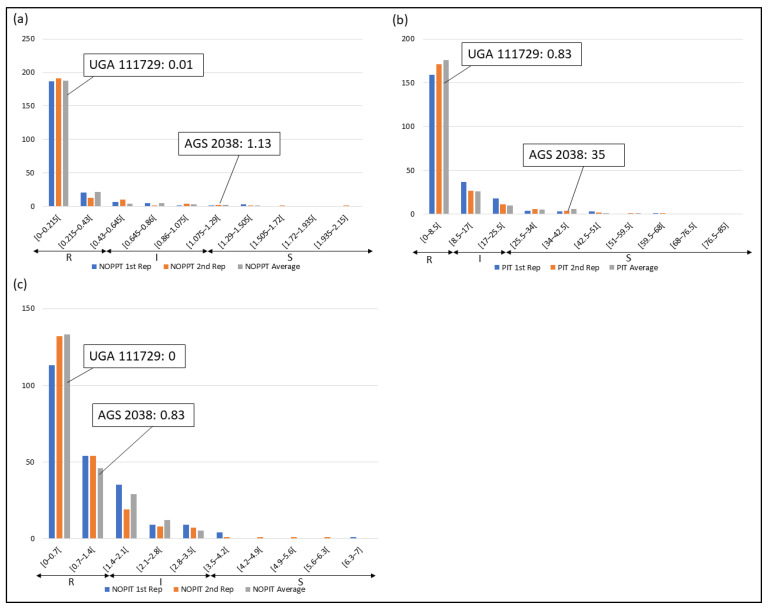
Frequency distributions for Plains. (**a**) Number of pupae/larvae per tiller (NOPPT), (**b**) percent infested tillers (PIT) per sample, and (**c**) number of pupae/larvae per infested tiller per sample in 2021 in Plains. Arrows indicate the scale for RILs that are resistant (R—NOPPT: 0–0.56, PIT: 0–10%, NOPIT: 0–1.81), intermediate in resistance (I—NOPPT: 0.56–1.13, PIT: 0–20%, NOPIT: 1.81–3.61), and susceptible (S—NOPPT: >1.13, PIT: >20%, NOPIT: >3.61) for each trait. Thresholds for each trait are included in parentheses.

**Figure 2 genes-14-01812-f002:**
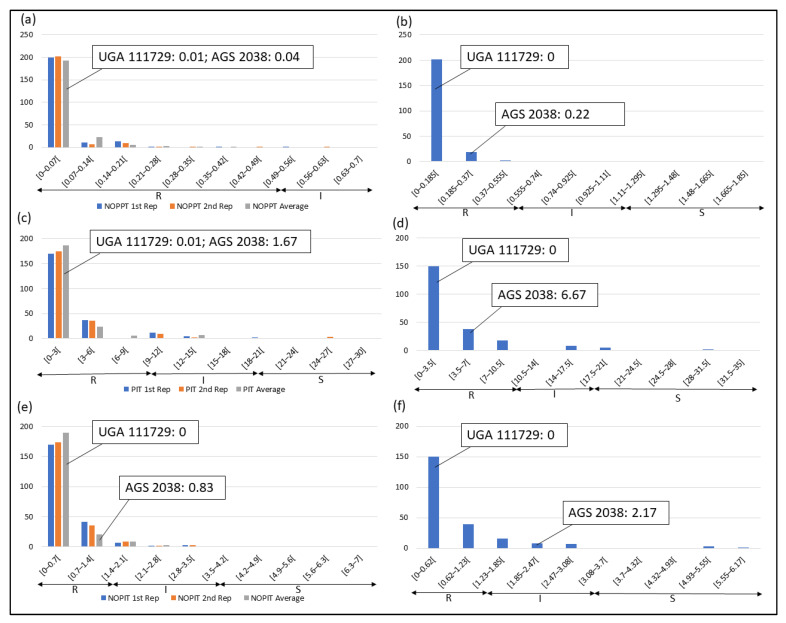
Frequency distributions for Williamson. (**a**) NOPPT in (**a**) 2021 and (**b**) 2022; (**c**) PIT in (**c**) 2021 and (**d**) 2022; (**e**) NOPIT in (**e**) in 2021 and (**f**) 2022. Arrows indicate the scale for RILs that are R (R—NOPPT: 0–0.56, PIT: 0–10%, NOPIT: 0–1.81), I (I—NOPPT: 0.56–1.13, PIT: 0–20%, NOPIT: 1.81–3.61), and S (S—NOPPT: >1.13, PIT: >20%, NOPIT: >3.61) for each trait. Thresholds for each trait are included in parentheses.

**Figure 3 genes-14-01812-f003:**
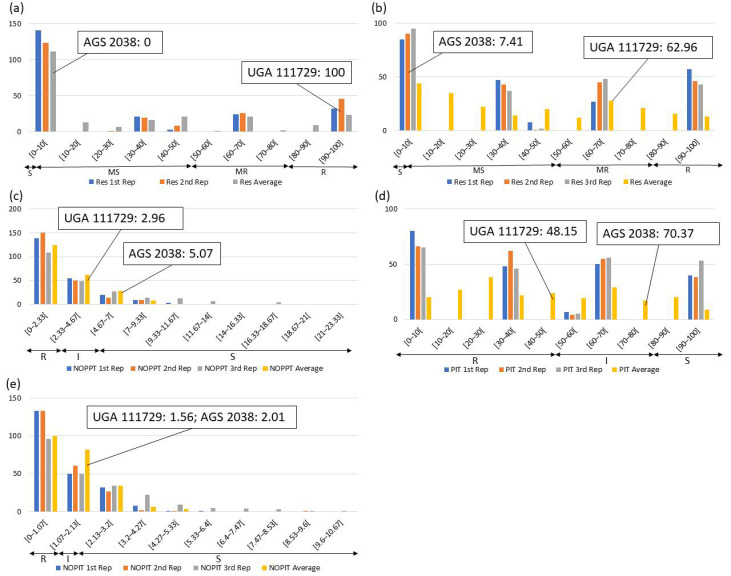
Frequency distributions for growth chamber experiments. (a) Percent resistant tillers (Res) according to visual scoring in (**a**) March 2019 and (**b**) October 2021; (**c**) NOPPT, (**d**) PIT, and (**e**) NOPIT in October 2021. Arrows indicate the scale for RILs that are R (R—Res: >80% for multi-gene ratio, >50% for single-gene ratio, NOPPT: 0–2.96, PIT: 0–48.15%, NOPIT: 0–1.56), moderately resistant (MR—Res: >50–80%), I (I—PIT: 48.15–70.37%, NOPPT: 2.96–5.07, NOPIT: 1.56–2.01), moderately susceptible (MS—Res: >0–50%), and S (S—Res: 0% for multi-gene ratio, 0–50% for single-gene ratio, NOPPT: >5.07, PIT: 70.37–100%, NOPIT: >2.01) for each trait. Thresholds for each trait are included in parentheses.

**Figure 4 genes-14-01812-f004:**
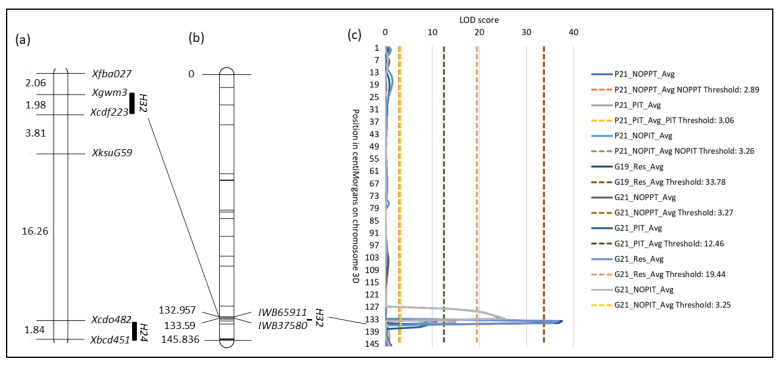
Location of *H32*. (**a**) Visualization of a composite linkage map of the long arm of chromosome 3D with RFLP and SSR marker data for *H24* and *H32*, respectively. Composite linkage map data were obtained from GrainGenes (https://wheat.pw.usda.gov/GG3/ (accessed on 15 March 2023)). Numbers on the left are cM distances between markers. (**b**) Linkage map of chromosome 3D and the location of SNP markers *IWB65911* and *IWB35780* that flank *H32*. Numbers on the left are the positions of the markers in cM. (**c**) QTL plot of LOD values of traits measured from field and growth chamber experiments.

**Figure 5 genes-14-01812-f005:**
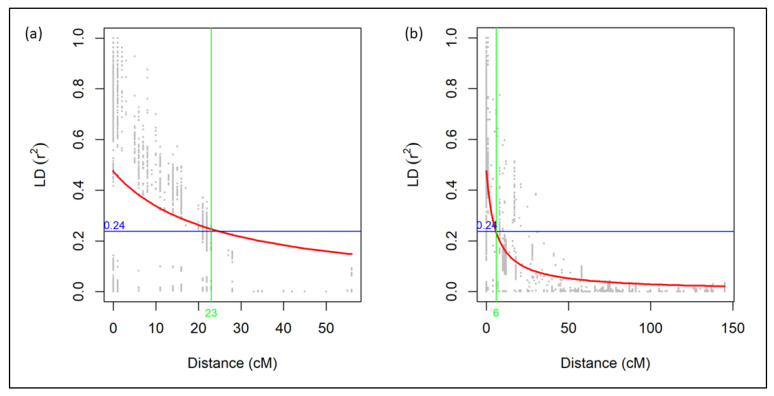
Linkage disequilibrium decay plots of linkage groups (**a**) 3A1 and (**b**) 3D of the UGA 111729 × AGS 2038 RIL population. The red line is the locally weighted polynomial regression (LOESS) curve; the blue line represents the *r^2^* critical value; and the green line represents the cM distance at which the LOESS curve and *r^2^* critical value intersect.

**Figure 6 genes-14-01812-f006:**
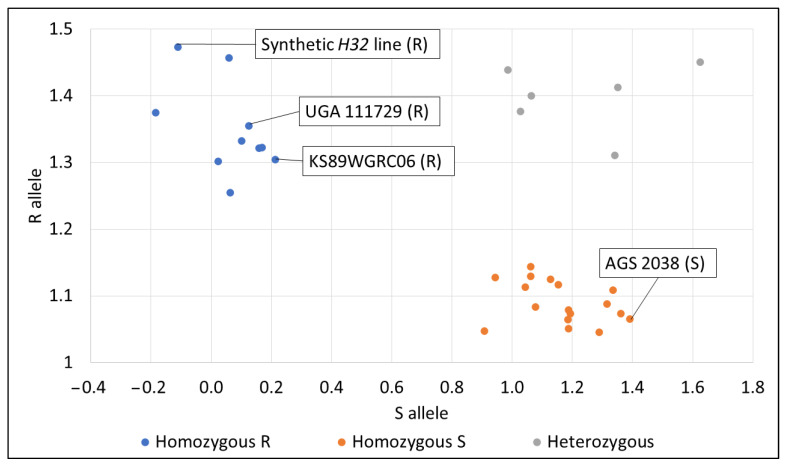
KASP marker validation results for lines with consistently resistant or susceptible Res phenotypes. R = resistant genotype, S = susceptible genotype.

**Table 1 genes-14-01812-t001:** Parental means of UGA 111729 × AGS 2038 recombinant inbred line (RIL) population for each trait and their standard errors (SE).

Trait	Location	Year	Abbreviation	UGA 111729		AGS 2038	
				Mean	SE	Mean	SE
PIT	Plains	2021	P21_PIT	0.83	0.83	35	5
	Williamson	2021	W21_PIT	0	0	1.67	0.83
		2022	W22_PIT	0	0	6.67	3.33
	Growth chamber	2021	G21_PIT	48.15	3.70	70.37	13.35
NOPPT	Plains	2021	P21_NOPPT	0.01	0.01	1.13	0.03
	Williamson	2021	W21_NOPPT	0	0	0.04	0.02
		2022	W22_NOPPT	0	0	0.22	0.15
	Growth chamber	2021	G21_NOPPT	2.96	1.58	5.07	1.84
NOPIT	Plains	2021	P21_NOPIT	0.17	0.17	3.61	0.66
	Williamson	2021	W21_NOPIT	0	0	0.83	0.44
		2022	W22_NOPIT	0	0	2.17	1.48
	Growth chamber	2021	G21_NOPIT	1.56	0.62	2.01	0.52
Res	Growth chamber	2019	G19_Res	100	0	0	0
		2021	G21_Res	62.96	7.41	7.41	7.41

**Table 2 genes-14-01812-t002:** Chi-square (X^2^) test of one gene or multiple gene segregation ratios for HF resistance in UGA 111729 × AGS 2038 RIL population for percent resistant tillers per cone for growth chamber results.

Trait	Year	Rep.	S		S Total	R		R Total	S:R = 1:1		S:MS:MR:R = 1:1:1:1	
			S	MS		MR	R		X^2^	*p*	X^2^	*p*
Res	2021	1	85	55	140	27	57	84	14	0	30.07	1.33 × 10^−6^
		2	90	44	134	45	46	91	8.22	0	27.04	5.79 × 10^−6^
		3	95	39	134	48	43	91	8.22	0	36.32	6.42 × 10^−8^
		Avg.	44	91	135	61	29	90	9	0	37.74	3.21 × 10^−8^
	2019	1	141	24	165	24	32	56	53.76	2.27 × 10^−13^	178.22	<2.2 × 10^−16^
		2	124	28	152	26	46	72	28.57	9.03 × 10^−8^	114.43	<2.2 × 10^−16^
		Avg.	112	57	169	24	32	56	56.75	4.95× 10^−14^	84.21	<2.2 × 10^−16^

Susceptible (S), Moderately susceptible (S), Moderately resistant (MR), Resistant (R), Replicate (R), Average (Avg.).

**Table 3 genes-14-01812-t003:** Chi-square (X^2^) test of one gene or multiple gene segregation ratios for HF resistance in UGA 111729 × AGS 2038 RIL population for PIT, NOPPT, and NOPIT for field results.

Trait	Year	Loc.	Rep.	R	I	S	Total	S:I:R 1:1:1	
								X^2^	*p*
PIT	2021	P	Rep 1	186	23	16	225	246.75	<2.2 × 10^−16^
			Rep 2	189	17	18	224	262.62	<2.2 × 10^−16^
			Average	193	13	19	225	278.72	<2.2 × 10^−16^
		W	Rep 1	219	6	0	225	414.96	<2.2 × 10^−16^
			Rep 2	218	2	4	224	412.75	<2.2 × 10^−16^
			Average	218	7	0	225	409.31	<2.2 × 10^−16^
		GC	Rep 1	128	57	40	225	58.11	2.41 × 10^−13^
			Rep 2	128	59	38	225	59.12	1.45 × 10^−13^
			Rep 3	111	61	53	225	26.35	1.90 × 10^−6^
			Average	131	48	46	225	62.75	2.37 × 10^−14^
	2022	W	Rep 1	206	14	4	224	347.18	<2.2 × 10^−16^
NOPPT	2021	P	Rep 1	215	6	4	225	392.03	<2.2 × 10^−16^
			Rep 2	211	10	3	224	373.72	<2.2 × 10^−16^
			Average	213	10	2	225	381.31	<2.2 × 10^−16^
		W	Rep 1	225	0	0	225	450	<2.2 × 10^−16^
			Rep 2	222	2	0	224	436.11	<2.2 × 10^−16^
			Average	225	0	0	225	450	<2.2 × 10^−16^
		GC	Rep 1	165	34	26	225	162.43	<2.2 × 10^−16^
			Rep 2	170	35	20	225	182	<2.2 × 10^−16^
			Rep 3	142	50	33	225	52.67	3.66 × 10^−12^
			Average	142	50	33	225	91.71	<2.2 × 10^−16^
	2022	W	Rep 1	222	1	2	225	432.19	<2.2 × 10^−16^
NOPIT	2021	P	Rep 1	187	33	5	225	256.11	<2.2 × 10^−16^
			Rep 2	195	25	4	224	293.85	<2.2 × 10^−16^
			Average	198	27	0	225	307.44	<2.2 × 10^−16^
		W	Rep 1	214	9	2	225	414.96	<2.2 × 10^−16^
			Rep 2	212	10	2	224	412.75	<2.2 × 10^−16^
			Average	218	6	1	225	409.31	<2.2 × 10^−16^
		GC	Rep 1	160	20	45	225	58.11	2.41 × 10^−13^
			Rep 2	168	24	33	225	173.52	<2.2 × 10^−16^
			Rep 3	118	27	80	225	55.71	8.01 × 10^−13^
			Average	148	28	49	225	109.52	<2.2 × 10^−16^
	2022	W	Rep 1	225	0	0	225	347.18	<2.2 × 10^−16^

S = susceptible, I = intermediate, R = resistant, Loc. = location, Rep. = replicate, *p* = *p*-value, P = Plains, W = Williamson, GC = growth chamber.

**Table 4 genes-14-01812-t004:** Broad-sense (*H^2^*) and narrow-sense (*h^2^*) heritability for all experiments conducted except the Williamson validation replicate from 2022.

Trait	Location	Year	Abbreviation	*H^2^*	*h^2^*
NOPPT	Plains	2021	P21_NOPPT	0.60	0.47
PIT			P21_PIT	0.64	0.52
NOPIT			P21_NOPIT	0.29	0.18
NOPPT	Williamson	2021	W21_NOPPT	0.12	0.03
PIT			W21_PIT	0.16	0.03
NOPIT			W21_NOPIT	0.19	0.06
Res	Growth chamber	2019	G19_Res	0.77	0.74
Res		2021	G21_Res	0.88	0.79
NOPPT			G21_NOPPT	0.47	0.33
PIT			G21_PIT	0.70	0.67
NOPIT			G21_NOPIT	0.39	0.27

**Table 5 genes-14-01812-t005:** Correlations between trait averages for HF response. Negative correlation is red, positive correlation is blue, and numbers on the top right half are *p*-values indicating statistical significance. Correlation coefficients are displayed in the bottom half. Darker red indicates a stronger negative correlation; lighter pink indicates a weaker negative correlation; lighter blue indicates a weaker positive correlation; and darker blue indicates a stronger positive correlation. Orange indicates statistically significant *p*-values.

	G19_Res_Avg	G21_Res_Avg	G21_NOPPT_Avg	G21_PIT_Avg	G21_NOPIT_Avg	P21_NOPPT_Avg	P21_PIT_Avg	P21_NOPIT_Avg	W21_NOPPT_Avg	W21_PIT_Avg	W21_NOPIT_Avg	W22_NOPPT_R1	W22_PIT_R1	W22_NOPIT_R1
**G19_Res_Avg**	1	*p < 0*.05	*p < 0*.05	*p < 0*.05	*p < 0*.05	*p < 0*.05	*p < 0*.05	*p < 0*.05	*p < 0*.05	*p < 0*.05	*p < 0*.05	*p < 0*.05	*p < 0*.05	*p < 0*.05
**G21_Res_Avg**	0.70	1	*p < 0*.05	*p < 0*.05	*p < 0*.05	*p < 0*.05	*p < 0*.05	*p < 0*.05	*p < 0*.05	*p < 0*.05	*p < 0*.05	*p < 0*.05	*p < 0*.05	*p < 0*.05
**G21_NOPPT_Avg**	−0.57	−0.74	1	*p < 0*.05	*p < 0*.05	*p < 0*.05	*p < 0*.05	*p < 0*.05	0.056	0.061	*p < 0*.05	*p < 0*.05	*p < 0*.05	*p < 0*.05
**G21_PIT_Avg**	−0.60	−0.84	0.83	1	*p < 0*.05	*p < 0*.05	*p < 0*.05	*p < 0*.05	0.088	*p < 0*.05	*p < 0*.05	*p < 0*.05	*p < 0*.05	*p < 0*.05
**G21_NOPIT_Avg**	−0.52	−0.67	0.90	0.70	1	*p < 0*.05	*p < 0*.05	*p < 0*.05	0.0615	*p < 0*.05	*p < 0*.05	*p < 0*.05	*p < 0*.05	*p < 0*.05
**P21_NOPPT_Avg**	−0.29	−0.33	0.23	0.34	0.20	1	*p < 0*.05	*p < 0*.05	*p < 0*.05	0.136	*p < 0*.05	0.135	*p < 0*.05	0.062
**P21_PIT_Avg**	−0.32	−0.38	0.26	0.37	0.23	0.96	1	*p < 0*.05	*p < 0*.05	*p < 0*.05	*p < 0*.05	0.092	*p < 0*.05	*p < 0*.05
**P21_NOPIT_Avg**	−0.31	−0.32	0.25	0.33	0.22	0.74	0.70	1	*p < 0*.05	0.064	*p < 0*.05	0.301	0.245	0.115
**W21_NOPPT_Avg**	−0.15	−0.17	0.13	0.12	0.12	0.09	0.16	0.13	1	*p < 0*.05	*p < 0*.05	*p < 0*.05	*p < 0*.05	0.094
**W21_PIT_Avg**	−0.16	−0.17	0.12	0.13	0.13	0.06	0.10	0.10	0.88	1	*p < 0*.05	*p < 0*.05	*p < 0*.05	*p < 0*.05
**W21_NOPIT_Avg**	−0.21	−0.24	0.21	0.17	0.20	0.13	0.22	0.14	0.80	0.67	1	*p < 0*.05	*p < 0*.05	0.077
**W22_NOPPT_R1**	−0.16	−0.20	0.16	0.16	0.18	0.12	0.13	0.09	0.16	0.25	0.15	1	*p < 0*.05	*p < 0*.05
**W22_PIT_R1**	−0.23	−0.26	0.22	0.22	0.22	0.19	0.21	0.10	0.21	0.33	0.20	0.79	1	*p < 0*.05
**W22_NOPIT_R1**	−0.16	−0.20	0.20	0.18	0.22	0.17	0.18	0.15	0.14	0.18	0.14	0.76	0.70	1

G19 = growth chamber 2019, G21 = growth chamber 2021, P21 = Plains 2021, W21 = Williamson 2021, W22 = Williamson 2022, Avg = Avg., R1 = Replicate 1. W22 only had one replicate.

**Table 6 genes-14-01812-t006:** Significant QTL results from the linkage groups 3A1, 3D, and 6B2.

QTL	Trait Abbreviation	Chr.	Marker at Peak/cM	Marker Interval/cM	LOD	R^2^	Add.
*QHf.ga.srww.3DL*	P21_NOPPT_R1	3D	*IWB65911*/132.957	*IWB26378*/126.977–*IWB19391*/136.836	4.02–37.55	7.18–67.48	−29.23–22.16
	P21_NOPPT_R2						
	P21_NOPPT_Avg						
	P21_PIT_R1						
	P21_PIT_R2						
	P21_PIT_Avg						
	P21_NOPIT_R1						
	P21_NOPIT_R2						
	P21_NOPIT_Avg						
	G19_Res_R2						
	G19_Res_Avg						
	G21_NOPPT_R1						
	G21_NOPPT_R2						
	G21_NOPPT_R3						
	G21_NOPPT_Avg						
	G21_PIT_R1						
	G21_PIT_R2						
	G21_PIT_R3						
	G21_PIT_Avg						
	G21_Res_R1						
	G21_Res_R2						
	G21_Res_R3						
	G21_Res_Avg						
	G21_NOPIT_R1						
	G21_NOPIT_R2						
	G21_NOPIT_R3						
	G21_NOPIT_Avg						
*QHf.ga.srww.3A*	P21_NOPPT_Avg	3A1	*IWB14875*/22.285	*IWA6387*/1.643–*IWB14875*/22.285	2.91–4.12	2.72–5.91	−0.04–0.28
	G21_NOPIT_R1						
*QHf.ga.srww.6B*	G19_Res_R2	6B2	*IWB59262*/68.412	*IWB62788*/63.327–*IWB59262*/68.412	36.40	26.16	−30.43

Chr. = chromosome, Pos. = position, R^2^ = percent phenotypic variation explained, Int. (cM) = interval in centimorgans (cM), The following are logarithm of odds (LOD) thresholds used to indicate significant QTL: P21_NOPPT: 2.89, P21_PIT: 3.06, P21_NOPIT: 3.26, G19_Res: 33.78, G21_Res: 19.44, G21_NOPPT: 3.27, G21_PIT: 12.46, G21_NOPIT: 3.23, W21_NOPPT: 2.01, W21_PIT: 3.16, W21_NOPIT: 2.99, W22_NOPPT: 1.98, W22_PIT: 3.09, and W22_NOPIT: 2.94.

## Data Availability

The data presented in this study are available on request from the corresponding author.

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
