# Peer review of "Quantitative Trait Locus Analysis of Hessian Fly Resistance in Soft Red Winter Wheat"

_genes, 2023, doi:10.3390/genes14091812_

Round 1

Reviewer 1 Report

The authors have described the discovery of quantitative trait loci analysis of Hessian fly resistance in soft red winter wheat. The overall results of the survey are novel. The experiment design of this study is comprehensive, specific, and well-executed, providing important reference value for cloning the key genes responsible for Hessian fly resistance in soft red winter wheat. Additionally, the study offers useful molecular markers for molecular breeding. Here are some of my concerns:

1. Some of the pictures in the article are not very clear. Could you please provide high-definition versions of these images?

2. The article lacks a conclusive statement and does not effectively summarize the results.

3. In the analysis of phenotypic data in section 2.5, the demonstration of data processing was unclear, and it was not specified which formula was specifically used in the article.

Minor editing of English language required.

Reviewer 2 Report

This study is very interesting and the manuscript is overall well written. Just a few questions:

1.       Lines 29-31. If I understand correctly, the last sentence of the abstract is trying to say that a SNP marker, IWB65911, linked to resistance gene H32, has been found and validated, thus has the potential to be a diagnostic marker. Why was resistance gene H24 mentioned? I think one or two sentences explaining why H24 was evolved might be clearer than the figure.

2.       In the introduction section, I can see that the authors added several sentences/paragraphs according to Dr. Bahri’s comments, which made the section more solid and comprehensive. However, I think it needs an overall touch to make the text flow and the logic smoother (or maybe the version I’m reviewing is not the final version since I can see the co-authors’ comments?).

3.       Line 293. You mentioned that KASP markers have been developed. Since the SNP of interest, IWB65911, is one from the 9k/90k array, did you extract the marker information from http://www.polymarker.info/? The source should be mentioned or cited.

4.       In table s6, I think it should be mentioned that the FAM and HEX tags have been attached to the 5’ end of the forward primers.

5.       Figure 6. There are 5 samples on the scatter plot that were called homozygous S but did not group with the others. They seem to be more like “both alleles”. Could you explain what they are and why they segregate from the other homozygous S?

Reviewer 3 Report

Dear sir, the present paper 'Quantitative trait locus analysis of Hessian fly resistance in soft red winter wheat' is already adequated for publication in Genes. The English is good, and the results are clearly presented. The discussion is also ready for publication. I detected small flaws that are in the annotated version of the draft that is attached. Most Latin names in the literature are not in italic letter, which is a common mistake when referencing MDPI journals.

Best regards

Reviewer 4 Report

Dear Authors,

The topic is very interesting and the manuscript is very well written. All parts of the research are well written and the introduction, results and discussion are at a high level.

Despite the importance of the topic at the present time and the work and efforts done by researchers in the field, I have some comments:

·         In materials and methods part, page 4. Field experimental design, please mention the distance between rows, between plants, date of sowing in each season….etc.

·         Page 4, lines 173-174, the sentences are not clear? What do you mean by ‘’ fields were irrigated if necessary to assure adequate germination’’. It is not clear, please rewrite.

·         Page 9, the resolution of all figures is not high.

Best wishes
